health and disease and epidemiology, ecology

pathogen, $R_0$, *Bombus*, microparasite, wild flower strip, bumblebee

**Author for correspondence:**
Emily J. Bailes
e-mail: emilyjbailes@gmail.com

# Host density drives viral, but not trypanosome, transmission in a key pollinator

Emily J. Bailes[1,2], Judit Bagi[1,3], Jake Coltman[4], Michelle T. Fountain[5], Lena Wilfert[6] and Mark J. F. Brown[1]

[1]Department of Biological Sciences, Royal Holloway University of London, Bourne Building, Egham TW20 0EX, UK
[2]Department of Molecular Biology and Biotechnology, University of Sheffield, Firth Court, Sheffield S10 2TN, UK
[3]Ear Institute, University College London, 332 Gray's Inn Road, London WC1X 8EE, UK
[4]Expedia Group, Angel Building, 407 St John Street, London EC1V 4AD, UK
[5]NIAB EMR, New Road, East Malling, Kent ME19 6BJ, UK
[6]Institute of Evolutionary Ecology and Conservation Genomics, University of Ulm, 89069 Ulm, Germany

EJB, 0000-0001-6486-7058; LW, 0000-0002-6075-458X; MJFB, 0000-0002-8887-3628

Supplemental feeding of wildlife populations can locally increase the density of individuals, which may in turn impact disease dynamics. Flower strips are a widely used intervention in intensive agricultural systems to nutritionally support pollinators such as bees. Using a controlled experimental semi-field design, we asked how density impacts transmission of a virus and a trypanosome parasite in bumblebees. We manipulated bumblebee density by using different numbers of colonies within the same area of floral resource. In high-density compartments, slow bee paralysis virus was transmitted more quickly, resulting in higher prevalence and level of infection in bumblebee hosts. By contrast, there was no impact of density on the transmission of the trypanosome *Crithidia bombi*, which may reflect the ease with which this parasite is transmitted. These results suggest that agri-environment schemes such as flower strips, which are known to enhance the nutrition and survival of bumblebees, may also have negative impacts on pollinators through enhanced disease transmission. Future studies should assess how changing the design of these schemes could minimize disease transmission and thus maximise their health benefits to wild pollinators.

## 1. Background

Understanding the spread of disease is of fundamental importance in wildlife ecology [1,2]. As species that are the focus of conservation efforts usually have small and declining populations, they are particularly vulnerable to disease outbreaks, which can cause high levels of mortality. Emerging infectious diseases, where 'spillover' from large managed populations to small endangered populations can occur repeatedly, pose a particularly significant threat [2–5]. Consequently, an understanding of transmission dynamics within and between populations is key to enabling management of such disease outbreaks and thus preventing host population extinction [6–8]. For example, modelling of rabies transmission between packs of Ethiopian wolves enabled a successful vaccination programme, resulting in the survival of these critically endangered canids [9].

A key aspect of epidemiology for horizontally transmitted parasites is host density. Host density has long been used as a key component of theoretical models because of its role in influencing contact rates [10–12]. Such theoretical work has received support from empirical epidemiological studies. For example, in small-scale laboratory-based studies using *Daphnia*, host density influenced the likelihood of infection by protozoan parasites [13,14]. Large-scale studies of humans also suggest that population size and density determine the baseline

transmission potential of influenza in the USA and seasonal transmission dynamics of measles in West Africa [15,16]. In populations of field voles, where the transmission of cowpox has been extensively studied, recent work suggests that density-dependent transmission is at least partially responsible for patterns of disease transmission [17].

One area where the understanding of mechanisms behind disease transmission is particularly important is supplemental feeding of wildlife, which is a frequently used management intervention to help support declining populations [18]. However, such feeding can alter host behaviour and physiology in ways that could influence disease transmission. In particular, increased provisioning is often associated with host aggregation and increased contact rates [10,19]. Consequently, it is important to understand how indirectly manipulating host density affects parasite transmission, so that conservation strategies can be implemented without further detrimental effects on target or interacting species. In the case of supplemental feeding of birds, several studies indicate that supplemental feeding is associated with higher prevalence of disease [20,21], although the results of field studies are not always conclusive [22]. Overall, a meta-analysis of supplemental feeding of wildlife suggested that intentional supplemental feeding increases infection outcomes, especially in the case of recreational feeding (bird feeding or feeding to enhance the tourist experience) [19]. This result varied across parasite taxa, with infection outcomes of bacteria, helminths and viruses, but not protozoa, positively associated with increased recreational feeding [19]. Interestingly, while supplemental feeding either had no effect on or increased either host density or abundance, these differences did not relate directly to infection outcomes. Consequently, the mechanisms underlying this variation in whether host density alters disease transmission remain unclear.

One important case of supplemental feeding is the use of wildflower strips as a source of forage for flower visiting taxa in agricultural areas, which has been widely advocated as a strategy to mitigate habitat loss and improve pollinator populations [23–25]. Such schemes are incorporated as funded strategies under agri-environment schemes in the European Union (e.g. [26]), and elsewhere (e.g. [27]). These interventions have been shown to have a positive effect on insect abundance and diversity [23,24,28]. However, as in other cases of supplemental feeding, these resources can also cause local increases in pollinator density [29]. Flowers are an important site for the transmission of parasites within and between pollinator species [30–35]. However, we still know very little about how wildflower strips alter disease transmission between pollinators [36]. Given the important role of disease in pollinator declines [37], whether these schemes alter disease epidemiology remains a key question.

Here, we use a controlled experimental approach to ask how bumblebee nest density impacts disease transmission in bumblebees, as a first step towards understanding how supplemental feeding and the density increases it produces might alter disease dynamics in pollinators. More specifically, we tested whether the transmission of two common parasites, a virus (slow bee paralysis virus; SBPV) and a trypanosome (*Crithidia bombi*), differed under semi-field conditions between low and high densities of bumblebee colonies. Our results have important implications for future management strategies to improve wild bee populations on agricultural land.

## 2. Methods

### (a) Experimental organisms

#### (i) Bumblebees

Colonies of *Bombus terrestris audax* with 10–12 workers were obtained from Biobest (Belgium). Upon arrival, all colonies were determined to be free of common cellular parasites by phase-contrast microscopy following Rutrecht & Brown [38] and of SBPV by reverse transcription–polymerase chain reaction (RT–PCR) ([39]; see the electronic supplementary material). All workers were marked with numbered opalith tags (Graze, Germany and Thornes, UK) upon arrival, and new callows (newly emerged adults) were subsequently tagged within 1 day of emergence. Colonies were randomly allocated to the six polytunnel compartments (figure 1). Within each compartment, different coloured tags were used, so that colonies could be discriminated from each other.

#### (ii) Parasites and inoculation protocol

*Crithidia bombi* (hereafter referred to as *Crithidia*) is a common and abundant parasite of bumblebees [40,41] that is known to be transmitted via flowers [31,35]. *Crithidia* significantly reduces colony founding and queen fitness [42], and thus is likely to have significant impacts on bumblebee populations in the wild. *Crithidia* was isolated from the faeces of 12 naturally infected *B. terrestris* queens collected from Windsor Great Park, UK, and purified following the method of Martin et al. [43], following Cole [44]. Following 3 h of starvation, each donor colony was inoculated *per os* with 10 000 viable cells per worker in 10 µl of 44% w/w sugar water.

SBPV is an RNA virus that is found in both honeybees (*Apis mellifera*) and bumblebees in the wild, but is particularly prevalent in bumblebees [45]. The infection dynamics of SBPV in individual bumblebees have been well described, and the virus is known to exhibit context-dependent virulence in bumblebee workers [39], comparable to the effects of *Crithidia* [42]. SBPV donor colonies were created by inoculating each worker individually with SBPV *per os* with approximately $10^8$ virus particles (see the electronic supplementary material for details of inoculum) in 10 µl of 44% w/w sugar 0.5 M PBS, following 3 h of starvation.

### (b) Experimental design

To determine the transmission dynamics of SBPV and *Crithidia* under field realistic settings, we grew wild flowers (see the electronic supplementary material) in two large (8 × 24 m) polytunnels, located at NIAB EMR, Kent, UK, in 2017. The same set of flower species were present in all compartments. Polytunnels were covered with polythene while all plants were still in a vegetative state to prevent contamination of flowers with parasites from wild insects. Each polytunnel contained three 8 × 6.6 m compartments made of fine mesh (0.6 × 0.66 mm). Colonies were assigned randomly to the six compartments. To create different bee densities, half the compartments contained three colonies and half the compartments contained six colonies (figure 1). Within each compartment, one colony was assigned to be the SBPV donor colony and another as the *Crithidia* donor colony. All other colonies within a compartment were free of SBPV and *Crithidia* (recipient colonies). At the start of the experiment, recipient colonies were placed in their respective compartments and allowed to forage (schematic in the electronic supplementary material, figure S3). In parallel, donor colonies were inoculated as outlined below. Five days following the placement of the recipient colonies into their respective compartments, donor colonies were added and allowed to forage for 2 days while the recipient colonies were closed (and thus inaccessible to bees from the donor colonies). This 2-day period allowed the donor colonies to learn where their nest was located and to minimize the amount of drifting of bees between colonies. Following this, workers were

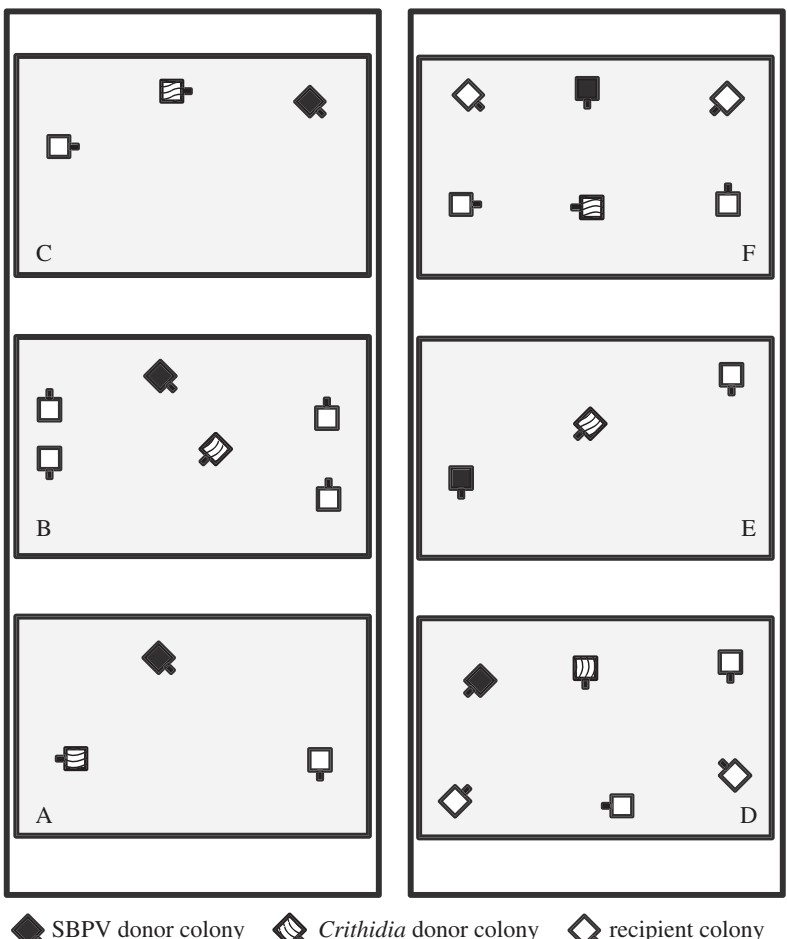

**Figure 1.** The layout of the colonies within the two polytunnels, each split into three compartments. Thick lines represent where bee excluding mesh (0.6 × 0.66 mm) was used, both around individual compartments (shaded in grey) and the entire polytunnel. SBPV donor colonies are shaded black, *Crithidia* donor colonies are striped and recipient colonies are shown in white. Compartments were 8 × 6.6 m.

destructively sampled from colonies and colony size equalized to approximately 16 workers (minimum = 10, maximum = 16; see the electronic supplementary material). The following morning, all colonies were opened to allow foraging. Both the donor and recipient colonies were open at the same time for a continuous period of 28 days, starting on 12 June 2017.

## (c) Sampling colonies for infection
### (i) Crithidia
Faecal samples were taken from individual workers representing 20% of each colony (minimum = three workers) every other day (electronic supplementary material, figure S3), including the night prior to opening all colonies. *Crithidia* infections can be identified in the faeces from 2 days after infection [46]. Owing to time constraints on faecal sampling, workers of a colony were screened for *Crithidia* until first detection within a colony, then removed from the sampling scheme. *Crithidia*-inoculated bees become infective within 2–5 days after exposure [46,47] and rapid spread of *Crithidia* has been observed within groups of workers in the laboratory [48]. Time to first detection can therefore be used as a proxy for transmission dynamics in this parasite. The faeces of individual workers were stored overnight at 4°C and then screened on a Nikon phase-contrast microscope at ×400 magnification. Samples were recorded for the absence or presence of transmission stages of *Crithidia* [47].

### (ii) Slow bee paralysis virus
To sample for SBPV, every fourth night, including the night prior to the introduction of donor colonies to the field, approximately

20% of the workers (minimum: two workers) were frozen in liquid nitrogen (electronic supplementary material, figure S3). SBPV viraemia peaks between 4 and 14 days post-inoculation [39]. For the first 12 days, bees were not sampled destructively if their colony had fewer than nine workers.

Colony size was estimated based on the number of workers in their natal colony on the night of sampling. Workers that had been directly inoculated in the donor colonies and workers less than 2 days old were excluded from the sampling scheme (but were included in the calculation of colony size). At the start (day 0) and the end of the experiment (day 28), in the donor colonies, a mixture of SBPV-inoculated and non-inoculated workers were sampled.

To screen individual workers for SBPV, they were bisected laterally and then RNA was extracted using the Tri-reagent based Direct-zol™ RNA MiniPrep kit (Zymo Research, CA, USA), which includes an on-column DNA digestion. Total complementary DNA (cDNA) was synthesized from 800 ng of RNA with random hexamers (Invitrogen) and oligodT (Primer Design) using M-MLV reverse transcriptase (Promega). RT–PCR was used to screen samples for the presence of viral RNA.

To reduce the likelihood of false positives and to derive a qualitative estimate of how much virus each sample contained, all experimental samples that tested positive for SBPV were tested twice. The band intensity in the second replicate reaction was then categorized as a strength from level 0 to 4, where 0, no virus, and 4, high virus. Over the entire duration of the experiment, the average percentage of SBPV-positive samples from the virus-inoculated colony in categories 1–4 was 34%, 25%, 27% and 14%, respectively. The methods for detecting SBPV in samples are described in full in the electronic supplementary material.

## (e) Assessments of drifting between colonies

To quantify the level of drifting between colonies, the location of workers in non-natal colonies was recorded every other night. Drifting was not a significant predictor of any measure of parasite transmission (see the electronic supplementary material).

## (f) Transect walks

To determine the density of foraging bumblebees within compartments, transect walks were undertaken. A path of approximately 25 m length was walked four times over a 20 min period, during which all workers identified feeding from flowers within approximately 1 m of the path were recorded, including their colony and unique identification (ID) number where possible. Transects were carried out every 4 days in a random order between compartments in immediate succession.

## (g) Flower density

Flower density was calculated every 4 days by recording the number of accessible floral units (those with open flowers) within five 0.5 m × 0.5 m quadrats haphazardly spaced across the compartment.

## (h) Statistical methods

### (i) Differences in bumblebee density

To test if we had successfully manipulated bumblebee density between compartments, a linear mixed model was fitted including time (continuous) and density (high or low) as fixed explanatory variables and compartment ID as a random factor. Bumblebee density (flowers per bee) was log transformed to meet the assumptions of normality and homoscedastic residuals. Differences in flower visitation rates were examined using a linear mixed model as above, flower visitation rate was also log transformed to meet the model assumptions.

### (ii) Time to first detection of *Crithidia* in a colony

To test if there was an effect of bumblebee density on the time taken for a bumblebee colony to become infected with *Crithidia*, a Cox proportional hazard model was fitted, with the response variable 'number of days until *Crithidia* detected within a colony'. Bee density (high or low) and colony treatment (SBPV-inoculated or recipient) were included as fixed factors and compartment ID was included as a random factor. All models met the assumption of proportional hazards.

### (iii) Slow bee paralysis virus transmission to recipient colonies

To test if bumblebee density had an effect on the likelihood of a worker testing positive for SBPV within a colony, a logistic regression model was fitted using a logit link function. Bumblebees from SBPV-inoculated colonies were excluded from the dataset. In one high-density and one low-density compartment, SBPV was not maintained in the SBPV-inoculated colony over the duration of the experiment (electronic supplementary material, figure S6), meaning that the treatment had failed in these compartments. Therefore, data points from these compartments were excluded from all subsequent models. Colony ID, nested within compartment ID, was included as a random factor in all models.

*Level of slow bee paralysis virus detected in samples.* To test if there was an effect of bumblebee density on the level of virus detected within a worker, the model was fitted using the same variables as in the binomial model above, but using a cumulative link mixed model with a flexible threshold. As there were very few data points in categories 3 and 4 (see the electronic supplementary material), the dataset was split into three categories: no virus detected; low levels of virus detected (level = 1); higher levels of virus detected (levels = 2–4). All models met the assumption of proportional odds.

All statistical analyses were carried out in R v. 3.4.1 [49]. The packages used are described in the electronic supplementary material and code can be accessed at https://gitlab.com/Jake-Coltman/bees-density-and-parasite-transmission. Models were selected by stepwise removal of predictors (initial and final models are given in the electronic supplementary material, tables S5–S12). *p*-values were calculated using log-likelihood ratio tests. All models were examined for their degree of multi-collinearity.

*Bayesian model of slow bee paralysis virus level.* To account for the multilevel nature of the data and potential autocorrelation of virus level, we also modelled the transmission of SBPV infections and their intensity at a colony level using a Bayesian random walk (code can be accessed at https://gitlab.com/Jake-Coltman/bees-density-and-parasite-transmission). The mean of the latent SBPV-level update step was modelled as a function of the level of infection in the donor colony and the density of the compartment. We first tested whether the additive impact of being in a high-density compartment on transmission rate, $\beta$, was greater than 0. The magnitude of the effect was evaluated by comparing posterior predictive samples generated using high-density and low-density time dynamics. This enabled us to assess how mean infection level within a given colony changed with respect to density and time.

The model was written in the probabilistic programming language Stan, via the pyStan library in Python 3.6 [50]. We took 5000 samples from five chains, with the first 2500 samples of each chain being used as a burn in. Samples were thinned such that only every fifth sample was kept.

# 3. Results

## (a) Bumblebee density was significantly higher in compartments with six versus three colonies

### (i) Bumblebee density

Over the duration of the experiment, there were $61 \pm 5$ bees (mean ± s.e.) in the low-density compartments compared to $111 \pm 10$ bees in the high-density compartments. The estimated number of flowers per compartment was not significantly different between treatments ($\chi_1^2 = 1.88$, $p = 0.17$), at $24\,000 \pm 2300$ and $20\,000 \pm 2300$ in the low- and high-density compartments, respectively. This resulted in 410 (95% confidence interval: 260–660) and 180 (120–290) flowers available per bee in the low- and high-density compartments, respectively; just over twice the numbers of flowers were available per bee in the high-density compartments ($\chi_1^2 = 8.19$, $p = 0.004$). This did not significantly change over the duration of the experiment ($\chi_1^2 = 2.74$, $p = 0.098$).

### (ii) Bumblebee visitation rates

The number of bees recorded foraging during the 20 minute observation periods was 14 (95% confidence interval: 10–19) in the low-density compartments and 16 (11–21) in the high-density compartments. There was no significant difference in the number of bees recorded foraging between the low- and high-density compartments during our seven observation periods ($\chi_1^2 = 0.82$, $p = 0.36$), but the number of bees observed did increase over time ($\chi_1^2 = 15.9$, $p < 0.001$).

## (b) The time until *Crithidia* infection was not significantly affected by bumblebee density

*Crithidia* was first detected in the faeces of worker bumblebees from uninfected recipient colonies between 6 and 14 days from

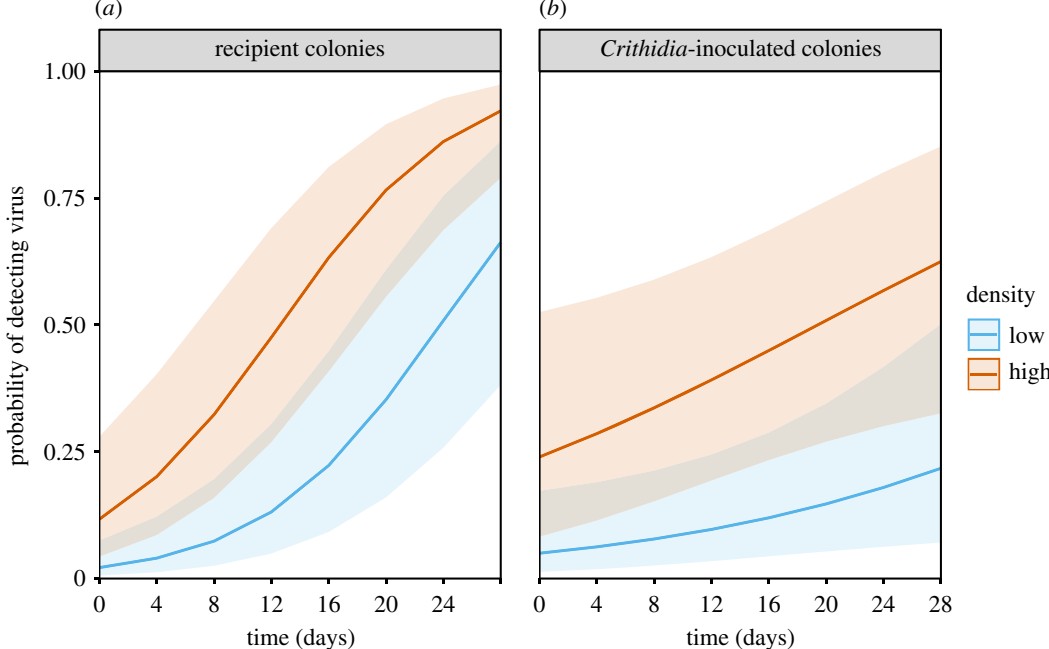

**Figure 2.** The predicted probability of detecting SBPV in a worker over the duration of the experiment for bees from recipient (*a*) and *Crithidia*-inoculated colonies (*b*), in high or low density treatments (colours given in right-hand legend), over the duration of the experiment; 95% confidence intervals are given by shaded areas. In the model; virus_detection = time × colony_treatment + density + (1|colony) + (1|compartment). Model estimates are given in table 1. (Online version in colour.)

the start of the experiment, with the prevalence at this point ranging from 0.2 to 1 (electronic supplementary material, figures S4 and S5). One colony was missed from sampling on day 10 but was infected at day 12; hence, we conducted analyses twice, with this colony coded as infected on either day 10 or day 12. As the analyses were similar, we report the analyses where the colony became infected on day 12 in the main text. Alternative analyses are presented in the electronic supplementary material.

In the Cox proportional hazard model, colony treatment (hazard ratio = 0.4 when treatment = SBPV-inoculated; $\chi_1^2 = 0.03$, $p = 0.9$) and bumblebee density (hazard ratio = 1.1 when density = high; $\chi_1^2 = 2.5$, $p = 0.12$) were both non-significant predictors of the time taken for a colony to become infected with *Crithidia*.

## (c) The detection of workers with slow bee paralysis virus is positively associated with bumblebee density

We identified replicative intermediates in a selection of SBPV-positive bees, including some from recipient colonies (see the electronic supplementary material, results ii). In accordance with previous infection assays in bumblebees, this suggests that SBPV detection in our samples is an indication of SBPV infection [39], although we cannot categorically exclude that some individuals tested positive without being actively infected. In the binomial logistic regression model, both treatment : time ($\chi_1^2 = 7.8$, $p = 0.0053$) and bee density ($\chi_1^2 = 4.0$, $p = 0.045$) were significant predictors of the likelihood of detecting virus in a bee (table 1 and figure 2). Bees in a high-density compartment were approximately six times more likely to become infected by SBPV than those in a low-density compartment. In addition, *Crithidia*-inoculated colonies (which were exposed to the SBPV-inoculated colonies 2 days before the control colonies) had a

**Table 1.** Fixed effect model estimates for the likelihood of detecting virus in a bee. (Predictor time, the time (4 day interval) at which the bee was sampled; treatment, whether the bee was from a 'non-inoculated' or '*Crithidia*-inoculated' colony; and density, whether the bee was from a 'low' or 'high' density compartment. Estimates for treatment are for '*Crithidia*-inoculated' colonies, and density for 'high' compartments, which are compared to the reference level of a non-inoculated recipient colony in a low-density compartment. *p*-values are not reported for time or treatment alone as their interaction is statistically significant.)

| predictor | estimate | s.e. | odds ratio | *p*-value |
|---|---|---|---|---|
| intercept | −3.814 | 0.664 | 0.022 | — |
| time : treatment | −0.404 | 0.144 | 0.67 | 0.005 |
| time | 0.641 | 0.094 | 1.90 | — |
| treatment | 0.868 | 0.595 | 2.38 | — |
| density | 1.792 | 0.693 | 6.00 | 0.045 |

correspondingly higher initial probability of detecting SBPV, but with a lower probability of SBPV being detected at the end of the experiment.

## (d) The level of slow bee paralysis virus in workers is positively associated with bumblebee density

SBPV-positive workers from colonies which were not experimentally infected had SBPV levels ranging from 1 to 4. While the majority of detections were level 1, there was overlap in the intensity level of SBPV in inoculated and non-inoculated compartments, especially towards the end of the experiment (electronic supplementary material, figure S7). Both treatment : time ($\chi_1^2 = 8.7$, $p = 0.0031$) and bee density ($\chi_1^2 = 4.4$, $p = 0.037$) were significant predictors of the level of virus in a bee (table 2). This suggests that in high-density

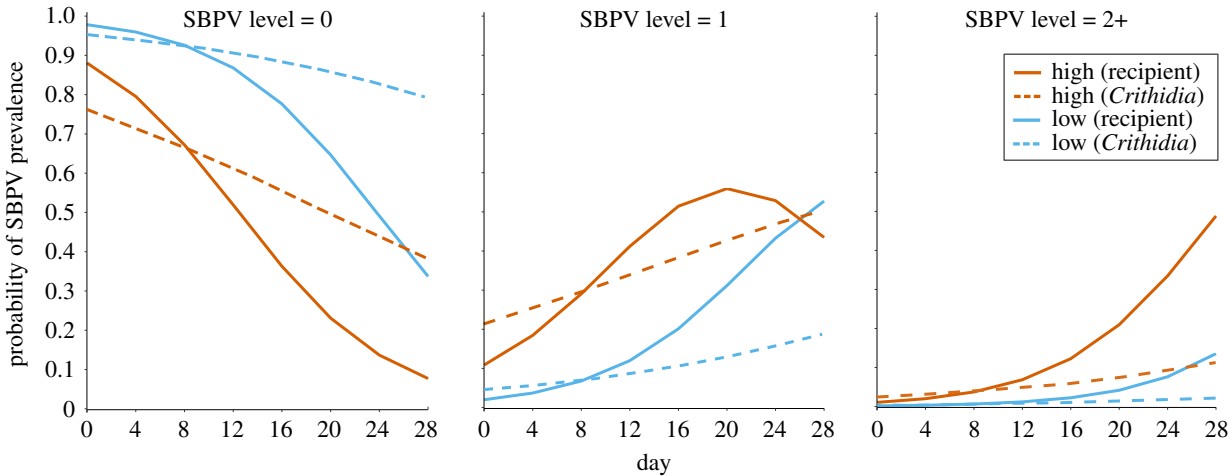

**Figure 3.** The predicted proportion of bumblebee workers in the categories of infection level 0 (no virus), 1 and 2+. Lines represent bumblebees from recipient (solid) and *Crithidia* donor colonies (dashed) at high and low colony density (colours given in right-hand legend) from the model; infection_level = time × treatment + density + (1|colony) + (1|compartment). Model estimates are given in table 2. (Online version in colour.)

compartments, individuals are significantly more likely to have higher levels of detectable SBPV (figure 3). These results are consistent with our Bayesian model. In 98.8% of samples, $\beta$, the additive impact of high density on transmission rate, was greater than 0 (electronic supplementary material, figure S8). This suggests that the transmission rate of SBPV was significantly higher in high-density compartments. In addition, based on the magnitude of $\beta$, we observed a substantial increase in latent SBPV-level using the high-density dynamics rather than the low-density dynamics. The median sample showed a 48% (inter-quartile range = 32–71%) increase in latent SBPV-level (electronic supplementary material, figure S9), suggesting that high nesting densities led to increases in the mean colony-level SBPV infection. This increase in latent SBPV-level corresponds to an increase in observed SBPV-level (i.e. a change from level 1 to level 2, etc.) in 47% of samples.

## 4. Discussion

Providing wildflower strips in agricultural areas has been widely advocated to conserve and promote pollinator populations [23,24]. However, very little is known about how the local increases in pollinator density that these strips produce [28,29] might influence disease transmission between individuals. Here, we show that bumblebee nest density can impact the transmission of disease between colonies using a controlled experimental approach. Interestingly, pathogen identity had a strong influence on disease transmission dynamics, with increased viral transmission being driven by higher density, in contrast with no impact of density on the transmission of a trypanosome parasite. In addition to impacts of density on transmission, our results suggest that increased nest density is positively associated with mean colony-level viral infection level.

Previous studies on the impacts of supplemental feeding on parasite transmission and prevalence have largely focused on vertebrates, and have identified a range of responses to how density changes, driven by such feeding, impact host–parasite dynamics (reviewed by Becker *et al.* [19]). Given fundamental differences in how host–parasite dynamics respond to nutritional supplementation in vertebrate versus invertebrate hosts

**Table 2.** Fixed effect model estimates for virus level detected in a sample. (Predictor time, the time (4 day interval) at which the bee was sampled; treatment, whether the bee was from a 'non-inoculated' or '*Crithidia*-inoculated' colony; density, whether the bee was from a 'low' or 'high' density compartment. Estimates for treatment are for '*Crithidia*-inoculated' colonies, and density for 'high' compartments, which are compared to the reference level of a non-inoculated recipient colony in a low-density compartment. *p*-values are not reported are time or treatment alone as their interaction is statistically significant.)

| predictor | estimate | s.e. | odds ratio | *p*-value |
|---|---|---|---|---|
| time : treatment | −0.404 | 0.144 | 0.668 | 0.003 |
| time | 0.641 | 0.094 | 1.899 | — |
| treatment | 0.868 | 0.595 | 2.382 | — |
| density | 1.792 | 0.693 | 6.002 | 0.037 |
| threshold: 1\|2 | 1.999 | 0.518 | — | — |
| threshold: 2\|3 | 4.529 | 0.593 | — | — |

[51], it was unclear how parasite transmission would respond to host density in our experimental system. Interestingly, host density enhanced viral transmission or vectoring, matching the results from an earlier meta-analysis [19]. This study is, to our knowledge, the first formal demonstration of SBPV transmission under semi-field conditions, and thus suggests that transmission of this virus, and perhaps others [45] (but see [36]), could be enhanced in agri-environment flower strips. An increase in SBPV prevalence could hypothetically also be seen if stressful conditions during the experiment were to activate undetected latent infections. While we cannot categorically refute this alternative hypothesis, the stringent molecular diagnostic test used means that such potential latent infections are likely to be very rare. Additionally, the prevalence data (electronic supplementary material, figure S6) show that two out of six inoculated donor colonies appear to be clearing the infection, with the recipient colonies in these departments not increasing in prevalence. This shows that, even if rare latent infections were present, they are highly unlikely to be the primary driver of the density-dependent prevalence patterns found in this experiment.

In contrast with SBPV transmission, we found no relationship between transmission of the trypanosome parasite and host density. Identifying a relationship between host density and parasite transmission requires an experimental design that brackets relevant changes in density. Consequently, it is possible that the lack of such an effect may be either because transmission had already peaked at the lowest density in our experiment, or our high-density compartments were not sufficiently populated. As our low-density treatment had a substantially higher density of nests than would be expected under natural conditions [52] this lends credence to the former explanation. Additional evidence in support of this view is that the average visitation rate in our experiment (approx. 0.0075 bees per metre of transect per minute of observation) was an order of magnitude higher than that seen for bumblebees in semi-natural and arable environments across the UK (average: approx. 0.00026 bees $m^{-1} min^{-1}$, max: approx. 0.00068 bees $m^{-1} min^{-1}$; [53]). However, when considering the number of flowers in our compartments, the average density of bees (approx. 0.0036 bees per metre of transect per 1000 flowers) is comparable to that seen on non-crop arable land (0.0068–0.008 bees $m^{-1}$ 1000 flowers$^{-1}$) and much lower than that seen at nectar-rich flower strips planted within arable land (0.025–0.077 bees $m^{-1}$ 1000 flowers$^{-1}$) reported by Carvell et al. [29]. If these are more relevant metrics for transmission, this would suggest that Crithidia transmission may already be at a plateau under agri-environment schemes. Clearly, further experiments, across a range of bumblebee nest densities, would be needed to investigate these possibilities. The manipulation of both nest density and the population size within those colonies would also help to disentangle whether the density of individuals or colonies within an area is a more important driver of disease transmission.

From a parasite perspective, whether or not increasing host density will have important effects on transmission rates will depend on the life history and interaction of the parasite with its host. Parasites with a low basic reproductive number $R_0$, showing low transmission rates, may benefit from increased host density, while those with very high transmission rates may show little increase in prevalence with a further increase in host density. For example, models by Bartlett et al. [54] have shown that for managed honeybees, increasing apiary size has marginal effects on the prevalence and transmission of established honeybee parasites with a very high $R_0$. By contrast, the increase in transmission rate and prevalence can be considerable for pathogens with lower base $R_0$ [54]. Mechanistically, the difference between our results for the two parasites may be a consequence of the inoculum required to produce a successful infection. Bumblebees shed sufficient Crithidia cells in a single defecation event to infect subsequent visiting workers [35,46,55], and this is reflected in a high prevalence in the wild of this parasite (e.g. [40]), corresponding to a high $R_0$. By contrast, bees will need to visit many flowers to achieve an infective dose of SBPV, as the infective dose is estimated to be approximately $10^8$ virus particles for the infection of B. terrestris with SBPV (E.J., J. Bagi, M.J.F. Brown 2019, unpublished data), whereas the viral load on a single flower has been quantified in the range of $10^2$–$10^6$ viral particles ([56]; E.J., J. Bagi, M.J.F. Brown 2019, unpublished data). Consequently, viral transmission probably occurs at a much lower rate, reflecting a low $R_0$, potentially explaining why only viral transmission responded to density in our experiment. As $R_0$ might vary within parasites

for different hosts [55], it would be interesting to see if the results of this study are constant across bumblebee species.

In a recent field study, Piot et al. [36] found that the prevalence of microparasites, including C. bombi, was higher in a focal bumblebee species (Bombus pascuorum) when wildflower strips were present in an otherwise florally depauperate landscape, but that there was no effect on viral prevalence; however, SBPV was not screened for in this study. At first sight, these results contrast with the patterns found in our controlled experimental trials, as they suggest that wildflower strips may lead to higher transmission of microparasites, but not viruses, under field conditions. This conclusion assumes that prevalence is a good proxy for inter-colony transmission, but this assumption is not necessarily valid—transmission of Crithidia occurs both within and between colonies, and as Piot et al. [36] did not determine the relatedness of the bees they sampled, it is impossible to tell how much of their prevalence derived from each of these transmission routes. In addition, wild bumblebees live in complex multi-species pollinator assemblages, which generate asymmetric patterns of flower sharing that are likely to drive both transmission [55] and vectoring [57] of microparasites. Two studies from Ireland [55] and Germany [58] suggest that host species differ in their importance as drivers of parasite prevalence in this system. Furthermore, there is growing evidence that secondary metabolites within pollen and nectar can mediate resistance to parasitic infection [59], and that pollen itself is necessary for the growth of some parasites [46,60]. Irrespective of these caveats, the contrast between our results for Crithidia transmission and those of Piot et al. [36] suggest that further controlled experiments at lower host density could be insightful for understanding the transmission dynamics of Crithidia under field conditions.

In addition to increasing the rate of viral transmission, our high-density treatment also increased the mean level of SBPV infections within colonies. While the relationship between infection level and virulence in SBPV has yet to be investigated, it is generally true that higher intensity infections in bee viruses have a higher impact on their hosts (e.g. [61,62]). Consequently, this result suggests that not only could changes in density increase transmission rates in flower strip agri-environment interventions but also that the impact of parasites and pathogens could be higher on individual bees and colonies.

Our experiment examined within-species transmission, but, as noted above, in the wild bumblebees live in complex multi-species assemblages of floral visitors. An increasing number of studies suggest that between-species transmission, in particular from managed honeybees, may be driving emergent diseases in wild pollinators [30,33,34,45,56,57,63–65]. As such, the next obvious step would be to conduct controlled semi-field trials to understand transmission dynamics of viruses between honeybees and bumblebees [34]. Ultimately, an understanding of the mechanism behind transmission dynamics should both help inform interpretation of well-designed field studies, and potentially enable the design of agri-environment interventions that nutritionally enhance bee health [28] while minimizing the potential for disease transmission.

## 5. Conclusion

Controlled semi-field experiments demonstrate the importance of density-dependent transmission for viruses in bumblebees. However, current agri-environment schemes designed to

support pollinator assemblages ignore their potential role for disease transmission among flower visitors. We suggest that future development of such schemes should take a more holistic, integrated approach that considers both nutrition and disease risk, to design conservation interventions that maximize pollinator health.

Ethics. Work complied with local ethical requirements.

Data accessibility. All data associated with this paper are available in the electronic supplementary material. Code for statistical analysis is available at https://gitlab.com/JakeColtman/bees-density-and-parasite-transmission/tree/master.

Authors' contribution. E.J.B., J.B., M.T.F., L.W. and M.J.F.B. conceived and designed the study; E.J.B. and J.B. carried out the field and laboratory work; E.J.B. and J.C. carried out the statistical analyses; E.J.B. and M.J.F.B. led the writing of the manuscript; all authors contributed critically to the writing of the manuscript and gave final approval for publication.

Competing interests. We declare we have no competing interests.

Funding. This work was funded by the BBSRC grant no. BB/N000668/1 awarded to M.J.F.B., grant no. BB/N000560/2 to M.T.F. and grant no. BB/N000625/1 to L.W.

Acknowledgements. We thank Robyn Manley for providing the initial SBPV inoculum for these experiments and for providing thoughtful input on the design and interpretation of this study, along with Vincent Doublet and Toby Doyle. We thank Lena Grinsted for statistical advice. The assistance provided by Graham Caspell, Tomas Bilcius and Adrian Harris onsite at NIAB EMR was greatly appreciated. We also thank members of the Brown and Leadbeater laboratories for their help and support with this experiment. The open access charge for this article was paid for by Royal Holloway University of London Library Services. We thank the editors and two anonymous reviewers for helpful feedback which improved the manuscript.

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
