## [Reviewer comments · Proceedings of the Royal Society B: Biological Sciences]

Review History

RSPB-2019-1969.R0 (Original submission)

Review form: Reviewer 1

Recommendation

Accept with minor revision (please list in comments)

Scientific importance: Is the manuscript an original and important contribution to its field?

Excellent

General interest: Is the paper of sufficient general interest?

Excellent

Quality of the paper: Is the overall quality of the paper suitable?

Good

Is the length of the paper justified?

Yes

Should the paper be seen by a specialist statistical reviewer?

No

Do you have any concerns about statistical analyses in this paper? If so, please specify them explicitly in your report.

Yes

It is a condition of publication that authors make their supporting data, code and materials available - either as supplementary material or hosted in an external repository. Please rate, if applicable, the supporting data on the following criteria.

Is it accessible?

Yes

Is it clear?

Yes

Is it adequate?

Yes

Do you have any ethical concerns with this paper?

No

Comments to the Author

See attached comments

Review form: Reviewer 2

Recommendation

Major revision is needed (please make suggestions in comments)

Scientific importance: Is the manuscript an original and important contribution to its field?

Excellent

General interest: Is the paper of sufficient general interest?

Excellent

Quality of the paper: Is the overall quality of the paper suitable?

Good

Is the length of the paper justified?

Yes

Should the paper be seen by a specialist statistical reviewer?

Yes

Do you have any concerns about statistical analyses in this paper? If so, please specify them explicitly in your report.

No

It is a condition of publication that authors make their supporting data, code and materials available - either as supplementary material or hosted in an external repository. Please rate, if applicable, the supporting data on the following criteria.

Is it accessible?

No

Is it clear?

Yes

Is it adequate?

Yes

Do you have any ethical concerns with this paper?

No

Comments to the Author

This manuscript describes the results of a cage experiment in which infected and non-infected bumble bees were allowed to forage on the same flowers and the presence of the pathogens was subsequently monitored in those same, originally uninfected bumble bees. The authors found that the viral pathogen increased in prevalence over time and, importantly, with host density whereas the gut pathogen (trypanosome: *Crithidia*) did not increase with host density. The experiment represents a considerable amount of effort. To place the study in broader context: there is worldwide concern over the reported decline of managed and non-managed bee species, with planting of flower strips within agricultural settings considered to be one option to reverse that trend. At the same time, there is growing awareness of the potential for pathogens to spill over from managed to non-managed bee species, which could lead to non-managed bee decline, and which could be exacerbated by the planting of flower strips (honeypot effect). The ms therefore addresses an important and outstanding issue in wildlife conservation.

The introduction is an excellent outline of the role of supplemental feeding of wildlife on parasite transmission and epidemiology; it is broad in taxa addressed and up-to-date. The methods are clearly described, statistical tests are appropriate, though the novel Bayesian modelling might need further explanation (or reviewing by a specialist in the field). Further justification for the pathogens employed would be helpful. Differences in host density (high versus low) were disappointingly small, which the authors acknowledge and discuss in the ms, but they call into question the relevance of the study to the real world. Results are presented in a visually appealing way, though I missed seeing data on absolute prevalence of *Crithidia* in the main body of the text with which to judge the interpretation of results. The discussion was well written, though necessarily focused on the caveats of the study, given that one pathogen (virus) followed the expected pattern (higher transmission at higher host density) whereas the other (trypanosome) did not.

Overall, I found the manuscript interesting and relevant, slightly tainted by the low replication (3 cages high host density, 3 cages low host density), small difference in host density and the insignificance of host density on transmission of the acknowledged trypanosome parasite *Crithidia*, though in praise of the experimental paradigm and clarity of the text. All comments are numbered below.

Supplementary data tables were complete. Supplementary Figures and descriptions of methods were missing (File S1 SEM contained two sentences, one terminated in mid-sentence). This may be a problem of the journal website.

Major comments (lines)

1. Host density. Though the experiment is well conceived and nicely set up, with 3 or 6 bumble bee colonies per cage (compartment), densities on flowers and frequencies of flower visitation were not very different between treatments (lines 292-7, difference n.s.). The authors rightly acknowledge this caveat in their discussion, and go to some lengths to interpret their results in the light of it (all credit to them). I accept that, a posteriori, there is not much that can be undertaken to rectify the issue. But it might strengthen the authors' arguments if they could relate their own flower visitation data to those from the field so as to provide more context for what has been undertaken experimentally. For example, I cannot tell if the host densities were around those expected in the field or whether both (high density and low density treatments) were extremely high (or extremely low). I acknowledge that such information will not rectify the lack of difference in flower visitation between the two treatments.

2. `Null control. Was a control cage used to test whether bumble bees in confinement (and a latent or unobserved infection with SBPV) exhibited SBPV? Though bees were checked for viral infection (which required sacrificing them), those introduced to the cages apparently uninfected were obviously not checked. Could their stress of confinement account for the pattern of increasing SBPV over time that was observed in the experiment?

3. (113-120) Did the *Crithidia* inoculum comprise viable trypanosomes?

4. *Crithidia*. This is a well-recognized pathogen of bumble bees so its use is well justified, even if no effect of host density on pathogen prevalence was seen by the authors. I see this treatment as some form of 'positive control' that the experimental set-up is working, but appreciate that the authors argue that prevalence did not change, possibly because of extremely high transmission in both treatment (high/low density). To help readers interpret this caveat of the authors, I think it important to add raw data on prevalence of *Crithidia* across the experiment rather than relegating it to a Supplementary Fig S2 (I assume the figure was in Fig S2, but the file was not available on the website for me to view).

5. Choice of pathogen. *Crithidia* is well justified. It's not clear why SBPV was chosen as only under bee stress has a mild impact of it been demonstrated on bumble bees (Manley et al. 2017 – as cited by the authors). (To be honest, I'm not sure what virus I would have selected as evidence for pathogenicity of viruses in *Bombus* is scant). Could it be that SBPV develops as a latent infection, or is even not even infective and is simply being vectored across individuals most of the time. If it were very robust to the environment, this could explain why it increased in prevalence across the experiment. This is, of course, not an argument against the ms as, even if just vectored, an important effect of host density on prevalence has been shown. Presentation of absolute titers of virus in bees (or some relative measure: e.g. difference between experimenter-infected and originally uninfected) in the main body of the text would go some way to help the reader interpret the intensity of 'infection' beyond the authors' current scalar of 0-4 (line 201) or, in statistical analyses, 0-2 (lines 249-251).

6. Transmission at flowers. The hypothesis implicit in the authors' argument is that pathogens were transmitted at flowers. Can the authors exclude the possibility that bees were moving into and out of multiple nests i.e. cage confinement led to transmission/vectoring? Given the size of the cages (compartments), the high number of bees and their (altered) behavior when in a cage, 'drifting' into and out of nests may have been frequent. The statement (lines 205-207) that drifting was not a significant predictor gives no idea about the degree of drifting. Did the authors search for pathogens on flowers, or were pathogen loads on flowers measured? Such information would

also strengthen the argument for flower-transmission. Did all compartments have the same range of flower species?

(Lines 188-189) Further support for the idea that transmission mainly occurs on flowers could be gotten from the comparison of non-inoculated bees from inoculated colonies and like-aged bees from recipient colonies. If the dynamics of infection were the same in the two groups, it might suggest that transmission at flowers was all-important. This idea, of course, assumes little drifting of individuals between colonies.

7. (173-174, 309-312) Recipient bees were screened till first detection in a colony of one bee. Could this all-nothing measure of Crithidia transmission have led to a lack of significance between host density and Crithidia transmission? Could the authors use all individuals in their analysis, with colony as a repeated measure to account for non-independence (as in the analysis of SBPV prevalence)?

8. Bayesian statistical model. Though well explained for a non-expert like myself, independent statistical advice might be warranted.

Minor comments (by line number)

141 'field' (?) Surely the experiment was undertaken in cages and so 'field' is not appropriate.

145 'sampled' This suggests animals were collected and killed, but I think you mean they were just 'checked' (or their faeces 'inspected')

151 'replicate significantly' I'm not sure what is meant because either the virus does or does not replicate (delete 'significantly').

189 I think 'colonies' should be 'bees' and 'they' should be 'colonies'

198-202 I could not access the supplementary materials so cannot check on the controls that were run to ensure high-quality RNA was extracted from individuals.

227 disambiguate: 'It with also'

342 is the subheading back-to-front i.e. 'level of SBPV' is associated with 'bumble bee density'?

Decision letter (RSPB-2019-1969.R0)

14-Oct-2019

Dear Dr Bailes:

Your manuscript has now been peer reviewed and the reviews have been assessed by an Associate Editor. The reviewers' comments (not including confidential comments to the Editor) and the comments from the Associate Editor are included at the end of this email for your reference. As you will see, the reviewers and the Editors have raised some concerns with your manuscript and we would like to invite you to revise your manuscript to address them.

We do not allow multiple rounds of revision so we urge you to make every effort to fully address all of the comments at this stage. If deemed necessary by the Associate Editor, your manuscript

will be sent back to one or more of the original reviewers for assessment. If the original reviewers are not available we may invite new reviewers. Please note that we cannot guarantee eventual acceptance of your manuscript at this stage.

Research ethics:

Use of animals and field studies:

Please submit a copy of your revised paper within three weeks. If we do not hear from you within this time your manuscript will be rejected. If you are unable to meet this deadline please let us know as soon as possible, as we may be able to grant a short extension.

Best wishes,
Dr Sasha Dall
mailto:proceedingsb@royalsociety.org

Associate Editor
Comments to Author:

Thank you for submitting your manuscript “Host density drives viral, but not trypanosome, transmission in a key pollinator” to Proceedings B. I have now received two reviews and evaluated to manuscript myself. We all agree that your manuscript is well-written and addresses an interesting topic. However, a number of points have been raised by the reviewers, which should all be addressed. Like reviewer 2, my main concern is the magnitude of the experiment and its applicability to natural scenarios. The reviewer offers some suggestions to help strengthen discussion around this limitation. Both reviewers also raise the need for more details in the statistical methods section, and reviewer 1 highlights several specific changes or additional tests that would improve the analysis. The reviewers also note several points for additional discussion, such as justification of the parasites chosen and the relationship between bee and colony density.

Reviewer(s)' Comments to Author:
Referee: 1

Comments to the Author(s)
See attached comments

Referee: 2

Comments to the Author(s)
This manuscript describes the results of a cage experiment in which infected and non-infected bumble bees were allowed to forage on the same flowers and the presence of the pathogens was

subsequently monitored in those same, originally uninfected bumble bees. The authors found that the viral pathogen increased in prevalence over time and, importantly, with host density whereas the gut pathogen (trypanosome: *Crithidia*) did not increase with host density. The experiment represents a considerable amount of effort. To place the study in broader context: there is worldwide concern over the reported decline of managed and non-managed bee species, with planting of flower strips within agricultural settings considered to be one option to reverse that trend. At the same time, there is growing awareness of the potential for pathogens to spill over from managed to non-managed bee species, which could lead to non-managed bee decline, and which could be exacerbated by the planting of flower strips (honeypot effect). The ms therefore addresses an important and outstanding issue in wildlife conservation.

The introduction is an excellent outline of the role of supplemental feeding of wildlife on parasite transmission and epidemiology; it is broad in taxa addressed and up-to-date. The methods are clearly described, statistical tests are appropriate, though the novel Bayesian modelling might need further explanation (or reviewing by a specialist in the field). Further justification for the pathogens employed would be helpful. Differences in host density (high versus low) were disappointingly small, which the authors acknowledge and discuss in the ms, but they call into question the relevance of the study to the real world. Results are presented in a visually appealing way, though I missed seeing data on absolute prevalence of *Crithidia* in the main body of the text with which to judge the interpretation of results. The discussion was well written, though necessarily focused on the caveats of the study, given that one pathogen (virus) followed the expected pattern (higher transmission at higher host density) whereas the other (trypanosome) did not.

Overall, I found the manuscript interesting and relevant, slightly tainted by the low replication (3 cages high host density, 3 cages low host density), small difference in host density and the insignificance of host density on transmission of the acknowledged trypanosome parasite *Crithidia*, though in praise of the experimental paradigm and clarity of the text. All comments are numbered below.

Supplementary data tables were complete. Supplementary Figures and descriptions of methods were missing (File S1 SEM contained two sentences, one terminated in mid-sentence). This may be a problem of the journal website.

Major comments (lines)

1. Host density. Though the experiment is well conceived and nicely set up, with 3 or 6 bumble bee colonies per cage (compartment), densities on flowers and frequencies of flower visitation were not very different between treatments (lines 292-7, difference n.s.). The authors rightly acknowledge this caveat in their discussion, and go to some lengths to interpret their results in the light of it (all credit to them). I accept that, a posteriori, there is not much that can be undertaken to rectify the issue. But it might strengthen the authors' arguments if they could relate their own flower visitation data to those from the field so as to provide more context for what has been undertaken experimentally. For example, I cannot tell if the host densities were around those expected in the field or whether both (high density and low density treatments) were extremely high (or extremely low). I acknowledge that such information will not rectify the lack of difference in flower visitation between the two treatments.
2. `Null control. Was a control cage used to test whether bumble bees in confinement (and a latent or unobserved infection with SBPV) exhibited SBPV? Though bees were checked for viral infection (which required sacrificing them), those introduced to the cages ass apparently

uninfected were obviously not checked. Could their stress of confinement account for the pattern of increasing SBPV over time that was observed in the experiment?

3. (113-120) Did the *Crithidia* inoculum comprise viable trypanosomes?

4. *Crithidia*. This is a well-recognized pathogen of bumble bees so its use is well justified, even if no effect of host density on pathogen prevalence was seen by the authors. I see this treatment as some form of 'positive control' that the experimental set-up is working, but appreciate that the authors argue that prevalence did not change, possibly because of extremely high transmission in both treatment (high/low density). To help readers interpret this caveat of the authors, I think it important to add raw data on prevalence of *Crithidia* across the experiment rather than relegating it to a Supplementary Fig S2 (I assume the figure was in Fig S2, but the file was not available on the website for me to view).

5. Choice of pathogen. *Crithidia* is well justified. It's not clear why SBPV was chosen as only under bee stress has a mild impact of it been demonstrated on bumble bees (Manley et al. 2017 – as cited by the authors). (To be honest, I'm not sure what virus I would have selected as evidence for pathogenicity of viruses in *Bombus* is scant). Could it be that SBPV develops as a latent infection, or is even not even infective and is simply being vectored across individuals most of the time. If it were very robust to the environment, this could explain why it increased in prevalence across the experiment. This is, of course, not an argument against the ms as, even if just vectored, an important effect of host density on prevalence has been shown. Presentation of absolute titers of virus in bees (or some relative measure: e.g. difference between experimenter-infected and originally uninfected) in the main body of the text would go some way to help the reader interpret the intensity of 'infection' beyond the authors' current scalar of 0-4 (line 201) or, in statistical analyses, 0-2 (lines 249-251).

6. Transmission at flowers. The hypothesis implicit in the authors' argument is that pathogens were transmitted at flowers. Can the authors exclude the possibility that bees were moving into and out of multiple nests i.e. cage confinement led to transmission/vectoring? Given the size of the cages (compartments), the high number of bees and their (altered) behavior when in a cage, 'drifting' into and out of nests may have been frequent. The statement (lines 205-207) that drifting was not a significant predictor gives no idea about the degree of drifting. Did the authors search for pathogens on flowers, or were pathogen loads on flowers measured? Such information would also strengthen the argument for flower-transmission. Did all compartments have the same range of flower species?

(Lines 188-189) Further support for the idea that transmission mainly occurs on flowers could be gotten from the comparison of non-inoculated bees from inoculated colonies and like-aged bees from recipient colonies. If the dynamics of infection were the same in the two groups, it might suggest that transmission at flowers was all-important. This idea, of course, assumes little drifting of individuals between colonies.

7. (173-174, 309-312) Recipient bees were screened till first detection in a colony of one bee. Could this all-nothing measure of *Crithidia* transmission have led to a lack of significance between host density and *Crithidia* transmission? Could the authors use all individuals in their analysis, with colony as a repeated measure to account for non-independence (as in the analysis of SBPV prevalence)?

8. Bayesian statistical model. Though well explained for a non-expert like myself, independent statistical advice might be warranted.

Minor comments (by line number)

141 'field' (?) Surely the experiment was undertaken in cages and so 'field' is not appropriate.

145 'sampled' This suggests animals were collected and killed, but I think you mean they were just 'checked' (or their faeces 'inspected')

151 'replicate significantly' I'm not sure what is meant because either the virus does or does not replicate (delete 'significantly?').

189 I think 'colonies' should be 'bees' and 'they' should be 'colonies'

198-202 I could not access the supplementary materials so cannot check on the controls that were run to ensure high-quality RNA was extracted from individuals.

227 disambiguate: 'It with also'

342 is the subheading back-to-front i.e. 'level of SBPV' is associated with 'bumble bee density'?

Decision letter (RSPB-2019-1969.R1)

02-Dec-2019

Dear Dr Bailes

I am pleased to inform you that your manuscript entitled "Host density drives viral, but not trypanosome, transmission in a key pollinator" has been accepted for publication in Proceedings B.

Open Access

Paper charges

Sincerely,

Dr Sasha Dall
